# Cell Therapies under Clinical Trials and Polarized Cell Therapies in Pre-Clinical Studies to Treat Ischemic Stroke and Neurological Diseases: A Literature Review

**DOI:** 10.3390/ijms21176194

**Published:** 2020-08-27

**Authors:** Masahiro Hatakeyama, Itaru Ninomiya, Yutaka Otsu, Kaoru Omae, Yasuko Kimura, Osamu Onodera, Masanori Fukushima, Takayoshi Shimohata, Masato Kanazawa

**Affiliations:** 1Department of Neurology, Brain Research Institute, Niigata University, Niigata 951-8585, Japan; hatakeyama.mas@gmail.com (M.H.); ininomiya@bri.niigata-u.ac.jp (I.N.); m08a015f@yahoo.co.jp (Y.O.); onodera@bri.niigata-u.ac.jp (O.O.); 2Translational Research Center for Medical Innovation, Foundation for Biomedical Research and Innovation at Kobe, Hougo 650-0047, Japan; k-omae@tri-kobe.org (K.O.); kimura@tri-kobe.org (Y.K.); 3Medical R&D, Fukushima & Partners, Nagoya, Aichi 458-0045, Japan; mfukushi@tri-kobe.org; 4Department of Neurology, Gifu University Graduate School of Medicine, Gifu 501-1194, Japan; shimo@gifu-u.ac.jp

**Keywords:** stroke, neurological disease, cell therapy, stem cell, microglia, mononuclear cell, PBMC, pleiotropic effects, polarization

## Abstract

Stroke remains a major cause of serious disability because the brain has a limited capacity to regenerate. In the last two decades, therapies for stroke have dramatically changed. However, half of the patients cannot achieve functional independence after treatment. Presently, cell-based therapies are being investigated to improve functional outcomes. This review aims to describe conventional cell therapies under clinical trial and outline the novel concept of polarized cell therapies based on protective cell phenotypes, which are currently in pre-clinical studies, to facilitate functional recovery after post-reperfusion treatment in patients with ischemic stroke. In particular, non-neuronal stem cells, such as bone marrow-derived mesenchymal stem/stromal cells and mononuclear cells, confer no risk of tumorigenesis and are safe because they do not induce rejection and allergy; they also pose no ethical issues. Therefore, recent studies have focused on them as a cell source for cell therapies. Some clinical trials have shown beneficial therapeutic effects of bone marrow-derived cells in this regard, whereas others have shown no such effects. Therefore, more clinical trials must be performed to reach a conclusion. Polarized microglia or peripheral blood mononuclear cells might provide promising therapeutic strategies after stroke because they have pleiotropic effects. In traumatic injuries and neurodegenerative diseases, astrocytes, neutrophils, and T cells were polarized to the protective phenotype in pre-clinical studies. As such, they might be useful therapeutic targets. Polarized cell therapies are gaining attention in the treatment of stroke and neurological diseases.

## 1. Introduction

Stroke remains a major cause of serious disability because the brain has a limited capacity to regenerate. In the last two decades, therapies for acute ischemic stroke (AIS) have changed dramatically, and the combination of mechanical thrombectomy (MT) and tissue plasminogen activator (tPA) administration recently showed a favorable outcome in the treatment of AIS. The therapeutic time window of tPA is 4.5 h after the onset of stroke, and that of MT is from 6 to 16 h with selection by perfusion imaging [1]. Furthermore, the therapeutic time window for this treatment has been extended, although only 5%–10% of patients with AIS are eligible for reperfusion treatment [2]. The reperfusion rate is almost 80% in patients with AIS treated with tPA and MT. However, half of the patients cannot achieve functional independence after treatment, and not all patients show an improved post-therapeutic outcome [3,4]. Therefore, additional treatments to improve the functional outcome are still needed in the subacute to chronic phases. Currently, many researchers are conducting clinical trials to investigate cell-based therapies that improve the functional outcome by acting as a neuronal replacement and/or a slow-release source of growth factors [5,6]. Several phase II clinical trials have been reported, and animal models have shown that recent approaches based upon the polarization hypothesis of cell status might improve angiogenesis, axonal outgrowth, and functional integration with neuronal networks [7,8,9,10]. The present review aims to describe the general benefits and mechanisms of cell therapies (in Section 3 and Section 4), conventional cell therapies under clinical trials (in Section 5 and Section 6), and advanced polarized cell therapies involving protective phenotypes in pre-clinical studies (in Section 7, Section 8 and Section 9) that facilitate functional recovery in patients with ischemic stroke in the subacute and chronic phases after post-reperfusion treatment.

## 2. Methods

A literature review was performed using PubMed as well as the National Institutes of Health clinical trial database (Clinicaltrials.gov). We searched articles published between April 1997 and July 2020 using the search terms “cell therapy”, “stroke”, “neurological disease”, “stem cell”, “microglia”, “mononuclear cell”, “microglial polarization”, “macrophage polarization”, “A1 and A2 astrocyte”, “N1 and N2 neutrophil”, and “T cell”.

## 3. General Benefits of Cell Therapies

“Single-target” therapies may be insufficient because ischemic cerebral injury involves several mechanisms [11]. In particular, tPA treatments can induce the following conditions: (1) direct ischemia/reperfusion injury, (2) tPA toxicity, (3) free-radical accumulation, (4) matrix metalloproteinase (MMP) activation, (5) inflammation, and (6) remodeling factor-mediated effects [12]. Each of these effects play important roles that change over time. For example, many researchers have investigated the therapeutic potential of vascular endothelial growth factor (VEGF). It drastically induces angiogenesis, neuroprotection, and possible axonal outgrowth in the chronic phase after stroke [13,14,15,16]. However, VEGF increases vascular permeability and cerebral edema in the acute phase of AIS, which can result in hemorrhage [17]. Moreover, systemic administration of VEGF induces vasodilation, increasing brain circulation and leading to hypotensive complications [18,19]. VEGF has important effects on endothelial cells, neurons, and glial cells, which together form the neurovascular unit [16,19]. VEGF studies have indicated that therapeutic approaches should target multiple mechanisms and various cell types to promote protection and recovery and that the timing of treatment is crucial. In this regard, cell-based therapies are ideal because they have multiple pleiotropic effects [11].

It is important to consider whether medications can cross the blood–brain barrier (BBB). According to Lipinski’s “rule of five”, five key physiochemical parameters affect passive diffusion trough the BBB: molecular weight (<500 Da), lipophilicity, polar surface area, hydrogen bonding, and charge [20]. It follows that small molecules cross the BBB by definition, while inflammatory and stem cells can cross the BBB via the action of adhesion receptor macrophage-1 antigen (cluster of differentiation [CD]11b) [9,21,22], stromal-derived factor-1 [23], or monocyte chemoattractant protein-1 [10,24,25]. The ability of these cells to cross the BBB can be exploited for therapeutic purposes.

## 4. Mechanisms of Cell-Based Therapies for Stroke

Cell-based therapies using bone marrow-derived mesenchymal stem/stromal cells (BM-MSCs) or bone marrow mononuclear cells (BM-MNCs) have pleiotropic mechanisms, so they may be ideal “multi-target” treatments for patients with stroke during the subacute and chronic phases [5,6]. Because stem cells can cross the BBB via adhesion molecules, they could likely be used to deliver target drugs into the brain. Additionally, these cells prompt functional recovery upon administration through the following three mechanisms (Figure 1) (Section 4.1, Section 4.2 and Section 4.3):

### 4.1. Direct Exchange of Damaged Neuronal Tissue and Neuronal Replacement by Administered Cells

Administered cells might differentiate into neuronal cells and form neuronal circuits with host neurons [7,26]. However, studies have shown that these neuronal replacement effects are limited and that other effects, known as bystander effects, play a greater role in cell therapies using non-neuronal bone marrow-derived stem cells [27,28].

### 4.2. Angiogenesis and Neuronal Remodeling

After cerebral ischemia, hypoxic tissues secrete VEGF, which promotes angiogenesis from the remaining vessels. In the central nervous system (CNS), microvessels and neurons are interrelated to form neurovascular units [29]. Thus, newborn vessels contribute to neuronal remodeling through various mechanisms after cerebral ischemia, and cell therapies contribute to neuronal remodeling by enhancing angiogenesis via several mechanisms [16,30,31]. Firstly, vessels supply oxygen and nutrients to the ischemic tissue. Secondly, VEGF secreted directly from the vessels promotes axonal outgrowth [14]. Laminin and β1 integrin expressed in endothelial cells may also promote axonal outgrowth [32]. Lastly, vessels contribute to endogenous neurogenesis [31,32,33,34]. In the adult brain, neurogenesis occurs in the rostral subventricular zone (SVZ) of the lateral ventricles and in the subgranular zone of the dentate gyrus after cerebral ischemia [35]. VEGF and fibroblast growth factor-2 released from endothelial cells may enhance neuronal stem cell (NSC) proliferation in these regions [33]. Following NSC proliferation, brain-derived neurotrophic factor (BDNF) secreted by endothelial cells promotes the migration of neuroblasts from the SVZ to the peri-infarct region [36]. In addition, neuroblasts express β1 integrin and adhere to laminin expressed by the vessels; subsequently, they migrate to the peri-infarct region using blood vessels as scaffolds [37].

### 4.3. Inhibition of Inflammatory Responses

Cerebral ischemia leads to the release of intracellular molecules such as nucleic acids and purine bodies from dying cells. These molecules, called damage-associated molecular patterns (DAMPs), act as danger signals and activate the innate immune system [38]. In the brain, DAMPs immediately activate resident microglia, which then secrete pro-inflammatory cytokines, such as tumor necrosis factor-α (TNF-α), interleukin (IL)-1β, IL-12, IL-23, and nitrogen monoxide (NO), all of which promote tissue damage [39]. In addition, circulating monocytes are recruited to the ischemic brain and differentiate into macrophages, which also become activated and promote tissue damage [38]. Similar to stem cells, anti-inflammatory cytokines, such as transforming growth factor-β (TGF-β), IL-4, and IL-10, may suppress inflammation-related tissue damage [40,41]; these cytokines are secreted by cells administered in cell therapies.

## 5. Cell Therapies Using BM-MSCS or BM-MNCS under Clinical Trials

Several types of cells can be used in cell therapies to treat ischemic stroke. In particular, non-NSCs, such as BM-MSCs and BM-MNCs, confer no risk of tumorigenesis, rejection, or allergy and pose no ethical issues. Thus, several recent clinical trials have used non-NSCs in cell therapies. BM-MSCs are positive for mesenchymal stem cell marker CD105 and negative for hematopoietic stem cell marker CD34. Conversely, BM-MNCs are negative for CD105 and positive for CD34. Cell surface markers help distinguish cell populations and characteristics (Table 1 and Table 2).

More than 10 years of both pre-clinical research and clinical investigation have evaluated the efficacy of BM-MSC therapy in ischemic stroke (Table 1). Honmou et al. reported the efficacy of cell therapy using BM-MSCs in the treatment of ischemic stroke in humans [42]. The study showed a reduction in infarct lesion volume and recovery of neurological function in patients treated with BM-MSCs. The effects of BM-MSC administration likely result from an increase in angiogenesis. In a phase I/Ⅱ a trial, surgical transplantation of modified BM-MSCs (SB623) via transient transfection with a vector encoding the human Notch1 intracellular domain led to functional recovery [43]. However, the phase Ⅱb trial showed no such functional recovery compared to the sham-operated group (ClinicalTrials.gov identifier: NCT02448641). Bone marrow-derived stromal cells have therapeutic potential against stroke. The efficiency and safety of autologous stromal cell transplantation have also been evaluated [44]. In particular, human cranial bone-derived MSCs (hcMSCs) express trophic factors such as BDNF and VEGF. The administration of hcMSCs promotes functional recovery after cerebral ischemia in a rat model [55]. Moreover, a clinical trial using hcMSCs is currently in progress in Japan.

BM-MNCs comprise several types of stem cells, so they are another promising source of cells to treat ischemic stroke. Indeed, several studies have reported the efficacy of cell therapies using BM-MNCs to treat ischemic stroke in humans [8,47,48,49,50,51,52,53,54] (Table 2). Although the mechanisms are still unclear, the administered BM-MNCs seem to promote endothelial cell proliferation and angiogenesis in the ischemic brain. These effects are followed by enhanced endogenous neurogenesis and functional recovery. BM-MNCs secrete trophic factors, such as VEGF, insulin-like growth factor-1, and small molecules, which lead to angiogenesis [27,47,48]. Delayed administration of BM-MNCs at 18.5 days after symptom onset may cause treatment inefficiency [49]. It follows that adequate timing of cell administration may be necessary to ensure favorable outcomes in the subacute to early chronic phase. The administration of human umbilical cord blood (HUCB)-MNCs also reduces infarct size and promotes functional recovery in ischemic rats [56]. The administration of HUCB-MNCs decreases the expression of pro-inflammatory cytokines and modulates the inflammatory response after cerebral ischemia [57]. However, HUCB-MNCs effects are clinically unknown because of the lack of clinical trials. In summary, clinical trials have mainly showed positive therapeutic improvement after administration of BM-MNCs (Table 2). More clinical trials should be performed to confirm these effects.

## 6. Other Cell Sources of Cell Therapies for Cerebral Ischemia

Multilineage-differentiating stress-enduring cells (Muse cells) are endogenous non-tumorigenic, pluripotent stem cells. Muse cells express the stem cell marker stage-specific embryonic antigen-3 (SSEA-3) and can generate all three germ layers. Muse cells can be collected from the bone marrow, adipose tissue, and dermal fibroblasts. After administration, Muse cells recognize an injured site, home into the injured tissue, differentiate spontaneously into tissue-compatible cells, and repair the tissue [58]. In cerebral ischemia, the therapeutic effects of Muse cell administration have been reported in an animal model, in which the administered Muse cells migrated to the injured tissue and differentiated into both neurons and oligodendrocytes. Thus, cell therapy using Muse cells is thought to function via neuronal replacement, unlike BM-MSC/BM-MNC cell therapy [45].

SSEA-1-expressing allogenic multipotent adult progenitor cells can be derived from bone marrow and are other candidates for cell therapy. The phase Ⅱ MultiStem^®^ Administration for Stroke Treatment and Enhanced Recovery Study (MASTERS) trial using multipotent adult progenitor cells showed that this cell therapy was safe within 24–48 h of symptom onset and that there was no functional difference between the cell therapy group and the placebo group at day 90 [46]. However, post-hoc analysis of the one-year results suggested that patients treated with multipotent adult progenitor cells may have continued to improve throughout the entire year, while the placebo-treated patients did not. Specifically, 28% of the patients in the cell therapy group had a modified Rankin scale ≤ 1 after one year, compared with only 13% of the patients in the placebo group, constituting a statistically significant difference. Based on this trial, the Treasure study is ongoing in Japan. A phase Ⅲ MASTERS-2 study (NCT02961504: ClinicalTrials.gov) is currently in the planning phase to evaluate the efficacy of this intervention in an earlier time window after stroke (<36 h).

The results (Table 1 and Table 2) described above may show promising clinical applications, even though the phase IIb trials of SB623 cells [43] and BM-MNCs [49,53] showed no functional recovery in the cell therapy group compared to the sham-operated group. However, the costs associated with these cell-based therapies amount to over $200,000 per patient. More clinical trials will confirm the cost-effectiveness and therapeutic effects of the treatments.

## 7. Cell Therapy Using Polarized Microglia in Pre-Clinical Studies

Although the cell therapies using BM-MSCs, BM-MNCs, and other cells described above are promising, it is difficult to obtain these cells for clinical applications for two reasons: (1) bone marrow aspiration is associated with increased risk in patients receiving antiplatelet therapy to prevent cerebral ischemia; (2) special equipment and long-term culture are required to prepare the cells for administration. Thus, cell therapy using cells adjusted by physiological reactions is preferable. Several next-generation cell sources have been investigated in pre-clinical studies, as have some cells that are more efficient in terms of the stimuli they induce. For example, microglia are a promising source for such cell therapy. They constitute 5%–10% of the total cell population within the normal brain and act as the first and main form of active immune defense intrinsic to the CNS [59]. Although the origin of microglia differs from that of monocytes/macrophages [60], their function is similar. Like macrophages, diversity and plasticity are hallmarks of microglial cells [39,61]. In response to pro-inflammatory or anti-inflammatory signals, both macrophages and microglia undergo M1-like (classical) or M2-like (alternative) activation [39,61]. Microglia present pro-inflammatory polarity and promote tissue injury in the acute phase of cerebral ischemia. Conversely, tissue protective microglia are detected from 12 h after cerebral ischemia and temporally increase after 1–3 days [62]. Tissue-protective microglia secrete trophic factors, such as VEGF, BDNF, platelet-derived growth factor (PDGF), and progranulin, as well as anti-inflammatory cytokines, such as TGF-β, IL-4, IL-10, and IL-13 [9,39,63]. Cell therapy using non-polarized microglia has no favorable effects [64]. However, a microglial polarity change is thought to enhance the therapeutic effect; this can be induced by mild ischemia [9], IL-4 and IL-13 [65], granulocyte macrophage colony-stimulating factor (GM-CSF) [8], or metformin [66] (Figure 2). Conversely, intravenous administration of IL-4 induced neutrophilic hyper-response and allergic reaction [67].

In the acute phase of cerebral ischemia, DAMPs activate microglia and peripheral mononuclear cells (PBMCs) to induce a pro-inflammatory phenotype. Several stimuli, such as mild ischemia, cytokines, and metformin induce microglia and PBMCs polarization to a tissue-protective phenotype, which may induce regeneration, including angiogenesis and axonal outgrowth, in the subacute to chronic phase of cerebral ischemia.

The ischemia-like stimulus oxygen–glucose deprivation (OGD) promotes microglial polarization [9]. Preconditioning by OGD promotes microglia acquisition of a tissue-protective polarity, enhancing the secretion of VEGF and TGF-β, as well as promoting MMP-9 activity [9,68]. The administration of microglia preconditioned by OGD (OGD-microglia) enhances angiogenesis and axonal outgrowth in the ischemic periphery in ischemic rats. It also improves neurological deficits after cerebral ischemia [9]. Microglia are highly plastic cells that can rapidly transit between different states. Indeed, they express both M1- and M2-like markers simultaneously [60,61], so the terminology and concept of M1- and M2-like microglia may be oversimplified [69]. Nonetheless, modulation of microglial polarization may have therapeutic application.

## 8. Cell Therapy Using Polarized PBMCs

OGD preconditioning is a simple method for preparing tissue-protective microglia, but it is challenging to obtain microglia from the adult human brain. Fortunately, cells with properties similar to those of microglia, such as monocytes and macrophages, exist among PBMCs. After cerebral ischemia, monocytes are recruited in the brain and differentiate into macrophages, which induce inflammation and promote tissue damage. Once recruited into the damaged tissue, macrophages lose their pro-inflammatory properties and secrete VEGF and TGF-β [70], indicating that, after cerebral ischemia, macrophages adopt an anti-inflammatory and tissue-protective phenotype.

The administration of macrophages with a tissue-protective polarity has therapeutic effects in patients with ischemic stroke [8]. However, the previous method for preparing tissue-protective macrophages requires long-term culture and medium supplemented with GM-CSF [8]. To allow clinical application, a simpler method for changing cell polarity must be developed.

OGD preconditioning is one of the simplest polarization methods. In one study, there were few stem cells among PBMCs, but their number increased after ischemic stroke. However, administering these cells did not induce beneficial effects [71]. Conversely, PBMCs preconditioned with OGD (OGD-PBMCs) secreted VEGF and TGF-β, similarly to microglia/macrophages (Figure 2) [10]. In addition, the percentage of SSEA-3-positive cells among PBMCs increased after preconditioning by OGD. After intra-arterial administration of OGD-PBMCs, the expression of VEGF and TGF-β, as well as the number of SSEA-3-positive cells in the ischemic periphery, increased in ischemic rats. Furthermore, the administration of OGD-PBMCs promoted angiogenesis and axonal outgrowth in the ischemic periphery, as well as functional recovery [10].

## 9. Polarization Strategies against Pathological Alterations

Researchers have investigated only one feature of pathologic alterations. For example, most researchers have evaluated either inflammatory or protective, anti-inflammatory aspects in stroke, traumatic injury, and neurodegeneration. However, recent studies have revealed that these targeted aspects are highly dynamic events that occur during stroke [9,10] (Figure 2) and other diseases.

In particular, reactive astrocytes were thought to form glial scars that inhibit axonal outgrowth. However, ablation of reactive astrocytes increased the number of CD45-positive activated microglia, preventing axonal outgrowth and diminishing functional recovery after spinal cord injury [72]. Relatedly, the activator of transcription 3, which is released by reactive astrocytes, may play a role in the repair of injured astrocytes [73]. Although astrogliosis has been considered irreversible, β1 integrin inhibition affected astrocytic polarization in on study, preventing glial scar formation and prompting axonal outgrowth [74]. Liddelow et al. found that neuroinflammation and ischemia induced two different types of glial fibrillary acidic protein-positive reactive astrocytes, termed neuroinflammatory (A1) and non-inflammatory (A2) (by analogy with M1/M2 macrophages) [75]. A1 astrocytes highly upregulate many classical complement cascades, especially C3, which are destructive to synapses. In contrast, A2 astrocytes express S100A10 and upregulate many neurotrophic factors, such as Clcf1 (neurotrophin 1) and pentraxin 3, and they show no characteristics of A1 astrocytes. Neuron-derived exosomes may promote functional behavioral recovery by suppressing the activation of M1-like microglia and A1 astrocytes in vivo and in vitro after traumatic spinal cord injuries [76]. Thus, A1 astrocytes, but not A2 astrocytes, have been induced using systemic injection of lipopolysaccharide. They have also been observed in active multiple sclerosis and Parkinson’s disease lesions in humans. Interestingly, 24 h after middle cerebral artery occlusion, both A1 and A2 astrocytes have been detected. In a Parkinson’s disease model, pro-inflammatory microglia mediated the conversion of astrocytes to a neurotoxic A1 phenotype [77]. Glucagon-like peptide-1 receptor agonist inhibited this conversion and prevented neuronal death, indicating that polarization of astrocytes is a promising therapeutic target.

In addition, knockout of toll-like receptor 4, which plays an important role in inflammation induction, polarized pro-inflammatory neutrophils (named N1) to a neuroprotective, alternative phenotype (N2) in ischemic stroke [78]. Regulatory T cells also play a role in the regulation of astrogliosis and neuronal recovery in the chronic phase of stroke. An increasing number of regulatory T cells among the T cell population might also be a therapeutic target [79]. Even in a tumor setting, adenovirus-transduced engineered macrophages showed M1-like polarization that played a critical role in the anti-tumor response [80]. These results were obtained by clustering data from advanced whole and single-cell RNA sequencing (RNAseq) techniques. The notion of polarization is a conceptual framework. However, because microglia, astrocytes, neutrophils, and stem cells have mixed and intermediate phenotypes, the terminology and concept of polarization may be complicated and oversimplified. Researchers should investigate how to polarize the protective phenotype, with particular focus on the adequate treatment timing. The “stem cell therapeutics as an emerging paradigm in stroke” (STEPS) group launched new standard guidelines to develop cell therapy [81]. According to these guidelines, the action mechanisms of cell therapy should be explored and defined in different animal models. To validate any relevant mechanism, a polarization hypothesis must be incorporated into the design of clinical trials as much as possible. However, distinct subtypes or phenotypes of cells may have different impacts at distinct phases of the CNS disease.

The concept of polarization involves thresholds rather than shading and/or proportionality. Classically, the ischemic penumbra was first defined by Astrup et al. as a zone of metabolically compromised tissue around a more densely affected ischemic core. The zone shows limited neuronal damage when regional cerebral blood flow (rCBF) is restored by rapid therapeutic intervention [82]. Although the precise mechanism of cell polarization is unknown, polarization concepts may be applied to the treatment of ischemic core and penumbra in the subacute to chronic phase, especially reperfusion therapies, which are established (Figure 3). Considering the double-faceted roles of cells and the advantages of their pleiotropic mechanisms for tissue repair, we suggest that polarization strategies could be applied to restore the brain parenchyma in pathological conditions and that they are easily accessible therapeutic treatments for patients with ischemic stroke and neurological diseases.

## 10. Conclusions

The administration of non-NSCs, such as BM-MSCs and BM-MNCs, as cell therapies is under clinical trial. Although cell therapies using BM-MSCs and BM-MNCs have promising clinical applications, it is unclear whether they will show any therapeutic improvement. A large number of clinical trials should be performed to confirm the real therapeutic effects of these treatments. The notion of cell polarization may be a conceptual framework. In fact, a polarized protective phenotype of astrocytes, neutrophils, microglia, T cell, and PBMCs may exist. Researchers should investigate how and when to induce the polarized, protective phenotype. Although future translational and clinical studies are required to support the idea of cell polarization, polarized cell therapies are gaining attention for the treatment of stroke and neurological diseases because they seem to have a protective function.

## Figures and Tables

**Figure 1 ijms-21-06194-f001:**
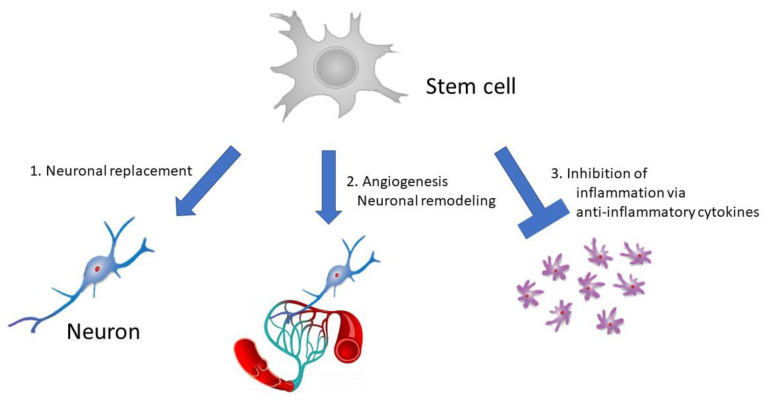
Schema of mechanisms of cell-based therapies for stroke.

**Figure 2 ijms-21-06194-f002:**
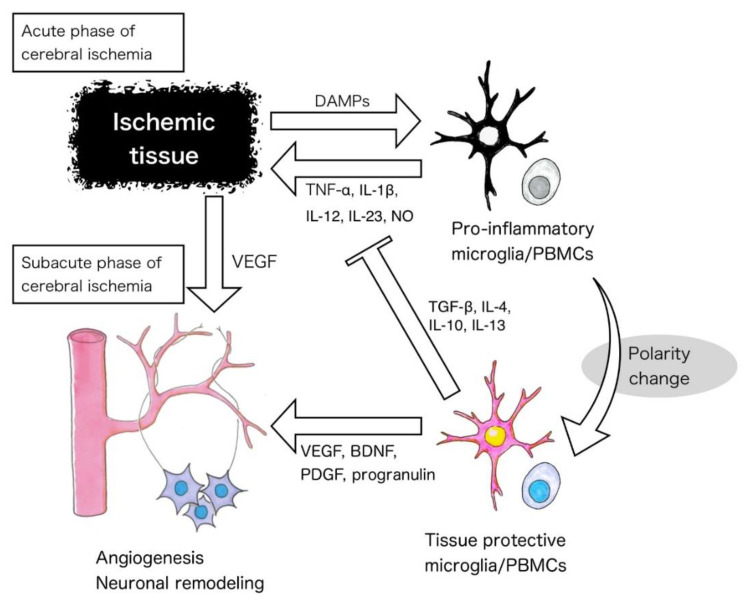
Mechanism of the therapeutic effects of microglia and peripheral mononuclear cells polarized to a tissue protective phenotype. DAMPs, damage-associated molecular patterns, TNF-α, tumor necrosis factor-α, IL, interleukin, PBMCs, peripheral blood mononuclear cells, TGF-β, transforming growth factor-β, VEGF, vascular endothelial growth factor, BDNF, brain-derived neurotrophic factor, PDGF, platelet-derived growth factor.

**Figure 3 ijms-21-06194-f003:**
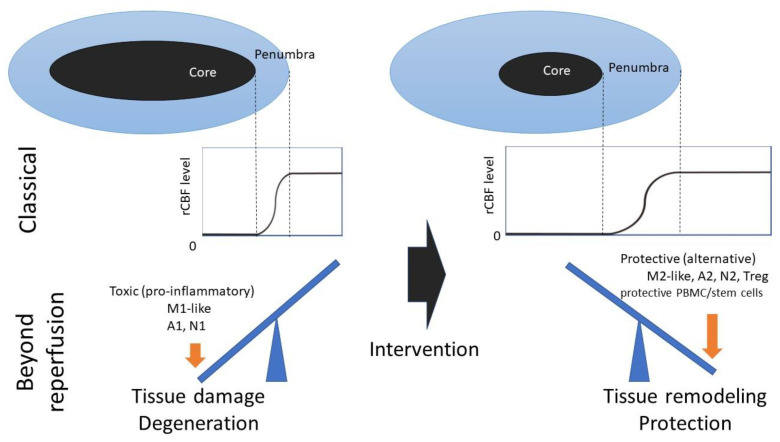
Scheme for polarization strategies. Treg, regulatory T cells. After reperfusion, microglia, astrocytes, neutrophils, regulatory T cells (Treg), PBMCs, M2-like polarized stem cells, A2 astrocytes, and N2 neutrophils may induce a protective, anti-inflammatory state via factors such as cytokines. Thus, mild ischemia and drugs may suppress inflammation and prompt tissue remodeling to salvage a lesion in patients with stroke.

**Table 1 ijms-21-06194-t001:** List of bone marrow-derived mesenchymal/stromal stem cell therapies under clinical trial.

Autologous Cells		Allogenic Cells	
Research Name Initiative	Cell Type (Product Name) Markers	Research Name Initiative	Cell Type (Product Name) Markers
INVEST-CI study Initiative: Sapporo Medical University [42]	Bone marrow-derived mesenchymal cells (STR01) CD34 (−) CD45 (−) CD105 (+) Ongoing	Initiative: SanBio company limited [43] NCT02448641	*Notch1*-transfected bone marrow-derived cells (SB623) CD29 (+) CD90 (+) CD105 (+) CD34 (−) CD45 (−) No beneficial effect
RAINBOW study Initiative: Hokkaido University [44]	Bone marrow-derived stromal cells (HUNS001) CD34 (−) CD45 (−) CD105 (+) Ongoing	Initiative: Tohoku University [45]	Bone marrow-derived cells CD34 (−) CD45 (−) SSEA-1 (+) Ongoing
		MASTERS study Initiative: Athersys, Inc., Healios K.K. [46]	Bone marrow-derived cells CD34 (−) CD45 (−) SSEA-1 (+) Ongoing

**Table 2 ijms-21-06194-t002:** List of bone marrow- and peripheral blood-derived mononuclear cell therapies under clinical trial.

Reference	ClinicalTrials.gov Identifier	Source	Study Design	Comments
Taguchi, et al. Stem Cell Dev 2015 [47], Kikuchi-Taura et al. Stroke 2020 [48]	NCT01028794	Autologous bone marrow mononuclear cell (CD34+)	Phase Ⅰ/Ⅱa	Improving outcome
Prasad, et al. Stroke 2014 [49]	NCT01501773	Autologous bone marrow mononuclear cell	Phase Ⅱ	No beneficial effect
Sharma, et al. Stroke Res Treat 2014 [50]	NCT02065778	Autologous bone marrow mononuclear cell	Phase Ⅰ	Improving outcome
	NCT00950521	Autologous peripheral blood stem cell CD34+)	Phase Ⅱ	No study results
Savitz et al. Ann Neurol 2011 [51] Vahidy et al. Stem Cells 2019 [52]	NCT00859014	Autologous bone marrow mononuclear cell	Phase Ⅰ	Safety
	NCT00473057	Autologous bone marrow cell	Phase Ⅰ	No study results
Ghali, et al. Front Neurol 2016 [53]	-	Autologous bone marrow cell	Open	No beneficial effect
Chernykh, et al. Cell Transplant 2016 [8]	-	Autologous blood mononuclear cell (CD14+)	Open	Improving outcome
Friedrich, et al. Cell Transplant 2012 [54]	-	Autologous bone marrow mononuclear cell	Open	Improving outcome

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
