# Peer review of "Cell Therapies under Clinical Trials and Polarized Cell Therapies in Pre-Clinical Studies to Treat Ischemic Stroke and Neurological Diseases: A Literature Review"

_ijms, 2020, doi:10.3390/ijms21176194_

Round 1

Reviewer 1 Report

The article presents a detailed study of techniques to intervene in cerebral stroke.

Although it may be an interesting article, it needs to be restructured, so that it can be understood.
First of all, it is not clear if it is a systematic review, or a study, so it would be necessary to modify the title.
As for the methodology, he has not cited any, making it difficult to validate the study.

On the other hand, it is necessary to restructure everything again, since there really is a very abstract article. some scheme or image would be appreciated to clarify the process followed.

The conclusions are not clear, and they affirm that they need clinical studies to support them.

Therefore, it is necessary to restructure the entire article, starting with the title, following with the abstract, and then with the structure of the article, since this makes it very difficult for the reader.

It is not really clear what is the message you want to convey, and it does not provide knowledge, since it probes many procedures, without clarifying their validity. reason why the professionals have it very difficult to apply some after reading the article.
Specific treatment of strokes is necessary since it leaves many sequels, and I understand that the intention of the article is that, although it is not well structured and the reader is not allowed to draw conclusions.

Author Response

1) Although it may be an interesting article, it needs to be restructured, so that it can be understood.
First of all, it is not clear if it is a systematic review, or a study, so it would be necessary to modify the title.
As for the methodology, he has not cited any, making it difficult to validate the study.

Reply:

Thank you for your comments. We have restructured the review to be concise and have stated explicitly whether the results discussed were from clinical trials or animal studies. Moreover, we discussed (1) the general benefits and mechanisms of cell therapies, (2) conventional cell therapies currently under clinical trial, and (3) advanced cell therapies using polarized, protective phenotypes, which are in pre-clinical studies to facilitate functional recovery in patients with ischemic stroke. Accordingly, we have revised Table 1 and added Table 2 about mononuclear cell therapies under clinical trial to discuss. We have also changed the title of the paper, revised the abstract and introduction, added supportive references, and revised the perspective discussion and conclusion.

(title)

Cell therapies under clinical trials and polarized cell therapies in the pre-clinical studies to treat ischemic stroke and neurological diseases: A literature review

(page 2, introduction)

“Currently, many researchers are conducting clinical trials to investigate cell-based therapies that improve functional outcome by acting as a neuronal replacement and/or a slow-release source of growth factors [5,6]. Several phase Ⅱ clinical trials have been reported, and animal models have shown that recent approaches based upon the polarization hypothesis of cell status might improve angiogenesis, axonal outgrowth, and functional integration with neuronal networks  [7,8,9,10]. The present review aimed to describe the general benefits and mechanisms of cell therapies (in the section 2,3),  conventional cell therapies under clinical trials (in the section 4,5) and 3. advanced polarized cell therapies involving protective phenotypes in pre-clinical studies (in the section 6,7,8) that facilitate  functional recovery in patients with ischemic stroke in the subacute and chronic phases after post-reperfusion treatment.”

On the other hand, it is necessary to restructure everything again, since there really is a very abstract article. some scheme or image would be appreciated to clarify the process followed.

Reply:

Thank you for your comments. Accordingly, we have revised Table 1 by adding “ongoing” or “no beneficial effects” to be concise. We added Figure 1 about schema of mechanism of mechanisms of cell-based therapies for stroke and Table 2 about mononuclear cell therapies under clinical trial to be concise and discuss.

The conclusions are not clear, and they affirm that they need clinical studies to support them.

Reply:

Thank you for your comments. Accordingly, we have revised conclusion.

(Conclusion)

The administration of non-NSCs, such as BM-MSCs and BM-MNCs, is under clinical trial and it is unclear whether it will show any therapeutic improvement. A large number of clinical trials should be performed to confirm the real therapeutic effects of these treatments. The notion of cell polarization may be a conceptual framework. Researchers should investigate how and when to polarize the protective phenotype. Although future translational and clinical studies are required to support the idea of cell polarization, polarized cell therapies are gaining attention in the treatment of stroke and neurological diseases because they seem to have a protective function.

Therefore, it is necessary to restructure the entire article, starting with the title, following with the abstract, and then with the structure of the article, since this makes it very difficult for the reader.

Reply:

Thank you for your comments. We have also changed the title of the paper, revised the abstract, added supportive references (#1, #50, #51, #53, #71, #76, and #81), and revised the perspective discussion and conclusion.

(Page 9 to 10)

However, because microglia, astrocytes, neutrophils, and stem cells have mixed and intermediate phenotypes, the terminology and conception of polarization may be complicated and oversimplified. Researchers should investigate how to polarize the protective phenotype, with particular focus on the adequate treatment timing. The stem cell therapeutics as an emerging paradigm in stroke (STEPS) group launched new standard guidelines to develop cell therapy [81]. According to these guidelines, the action mechanisms of cell therapy should be explored and defined in different animal models. To validate any relevant mechanism, a polarization hypothesis must be incorporated into the design of a clinical trials as much as possible. However, distinct subtypes or phenotypes of cells may have different impacts at distinct phases of CNS disease.

The concept of polarization involves thresholds rather than shading and/or proportionality. Classically, the ischemic penumbra was first defined by Astrup et al. as a zone of metabolically compromised tissue around the more densely affected ischemic core. The zone shows limited neuronal damage when regional cerebral blood flow (rCBF) is restored by rapid therapeutic intervention [82]. Although distinct mechanism of cell polarization is fully unknown, polarization concepts may be fitted into the ischemic core and penumbra in the subacute to chronic phase, especially reperfusion therapies are established presently (Figure 3). Considering the double-faceted roles of cells and the advantages of their pleiotropic mechanisms to tissue repair, we suggest that polarization strategies could be applied to restore brain parenchyma in pathological conditions, and that they are easily accessible therapeutic targets in ischemic stroke and neurological diseases.

Reviewer 2 Report

The manuscript reviews a topic of current interest but lacks in-depth analysis. It merely constitutes a fast overview of the field. Ideas and concepts are thrown in an irregular, shallow pattern. Sometimes are easy to follow, others appear not entirely finished. In summary, its contribution to the field is very limited.

Specific comments:

A more detailed explanation of cells used for therapy, their main biological/functional differences, differentiation of pre-clinical and clinical results, discussion of pros and cons, clarification of mainstream ideas…..etc is needed. The text is written in a sketchy manner. Ideas and data are delivered in such a way that it is not easy to get the message. The final message is very weak.

Minor: language requires extensive editing. Reading the text should be more fluid, sentences are arranged in awkward style at times.

Abstract: I guess the authors meant that “more than half of the patients do not achieve functional independence”

Author Response

1) A more detailed explanation of cells used for therapy, their main biological/functional differences, differentiation of pre-clinical and clinical results, discussion of pros and cons, clarification of mainstream ideas…..etc is needed. The text is written in a sketchy manner. Ideas and data are delivered in such a way that it is not easy to get the message. The final message is very weak.

Reply:

Thank you for your comments. We have restructured the review to be concise and have stated explicitly whether the results discussed were from clinical trials or animal studies. Accordingly, we have revised Table 1 by adding “ongoing” or “no beneficial effects” to be concise, as well as by distinguishing the biological differences of cell markers. We also added Table 2 about mononuclear cell therapies under clinical trial to discuss.

We added additional references (#1, #50, #51, #53, #71, #76, and #81) to support our ideas. In particular, we have discussed (1) the general benefits and mechanisms of cell therapies, (2) conventional cell therapies currently under clinical trial, and (3) advanced cell therapies using polarized, protective phenotypes, which are in pre-clinical studies to facilitate functional recovery in patients with ischemic stroke. We have also changed the title of the paper, revised the abstract and introduction, added supportive references, and revised the perspective discussion and conclusion.

(Introduction)

“Currently, many researchers are conducting clinical trials to investigate cell-based therapies that improve functional outcome by acting as a neuronal replacement and/or a slow-release source of growth factors [5,6]. Several phase Ⅱ clinical trials have been reported, and animal models have shown that recent approaches based upon the polarization hypothesis of cell status might improve angiogenesis, axonal outgrowth, and functional integration with neuronal networks  [7,8,9,10]. The present review aimed to describe the general benefits and mechanisms of cell therapies (in the section 2,3),  conventional cell therapies under clinical trials (in the section 4,5) and 3. advanced polarized cell therapies involving protective phenotypes in pre-clinical studies (in the section 6,7,8) that facilitate  functional recovery in patients with ischemic stroke in the subacute and chronic phases after post-reperfusion treatment.”

(Page 4)

BM-MSCs are positive for mesenchymal stem cell marker CD105 and negative for hematopoietic stem cell marker CD34. Conversely, BM-MNCs are negative for CD105 and positive for CD34. Cell surface markers help distinguish each cell population and characteristic (Table 1, Table 2).

(Page 6)

In cerebral ischemia, the therapeutic effects of Muse cell administration have been reported in an animal model, in which the administered Muse cells migrated to the injured tissue and differentiated into both neurons and oligodendrocytes. Thus, cell therapy using Muse cells is thought to function via neuronal replacement, unlike BM-MSC/BM-MNC cell therapy [57].

(Page 6)

The results (Table 1 and 2) described above may show promising clinical applications, even though the phase Ⅱb trials of SB623 cells [43] and BM-MNCs [48,53] showed no functional recovery compared to the sham-operated group. However, the costs associated with these cell-based therapies amount to are over $200,000 per patient. More clinical trials will confirm the cost-effectiveness and therapeutic effects of the treatments.

(Page 6)

However, the post-hoc analysis of the 1-year results suggested that patients treated with multipotent adult progenitor cells may have continued to improve throughout the entire year, while the placebo-treated patients did not. Specifically, 28% of the patients in the cell therapy group had a modified Rankin scale of ≤ 1 after 1 year, whereas only 13% of the patients in the placebo group had, constituting a statistically significant difference. Based on this trial, the Treasure study is ongoing in Japan. A phase Ⅲ MASTERS-2 study (NCT02961504: ClinicalTrials.gov) is currently in the planning phase to evaluate the efficacy of this intervention in an earlier time window after stroke (< 36 hours).

The results (Table 1 and 2) described above may show promising clinical applications, even though the phase Ⅱb trials of SB623 cells [43] and BM-MNCs [48,53] showed no functional recovery compared to the sham-operated group. However, the costs associated with these cell-based therapies amount to are over $200,000 per patient. More clinical trials will confirm the cost-effectiveness and therapeutic effects of the treatments.

  1. Cell therapy using polarized microglia in pre-clinical studies

Although the cell therapies using BM-MSCs, BM-MNCs, and other cells described above are promising, it is difficult to obtain these cells for clinical application for two reasons: (1) bone marrow aspiration is associated with increased risk in patients receiving antiplatelet therapy to prevent cerebral ischemia; (2) special equipment and long-term culture are required to prepare cells for administration. Thus, cell therapy using cells adjusted by physiological reactions is preferable. Several next-generation cell sources have been investigated in pre-clinical studies, as have some cells that are more efficient in terms of the stimuli they induce.

(Page 9 to 10)

However, because microglia, astrocytes, neutrophils, and stem cells have mixed and intermediate phenotypes, the terminology and conception of polarization may be complicated and oversimplified. Researchers should investigate how to polarize the protective phenotype, with particular focus on the adequate treatment timing. The stem cell therapeutics as an emerging paradigm in stroke (STEPS) group launched new standard guidelines to develop cell therapy [81]. According to these guidelines, the action mechanisms of cell therapy should be explored and defined in different animal models. To validate any relevant mechanism, a polarization hypothesis must be incorporated into the design of a clinical trials as much as possible. However, distinct subtypes or phenotypes of cells may have different impacts at distinct phases of CNS disease.

The concept of polarization involves thresholds rather than shading and/or proportionality. Classically, the ischemic penumbra was first defined by Astrup et al. as a zone of metabolically compromised tissue around the more densely affected ischemic core. The zone shows limited neuronal damage when regional cerebral blood flow (rCBF) is restored by rapid therapeutic intervention [82]. Although distinct mechanism of cell polarization is fully unknown, polarization concepts may be fitted into the ischemic core and penumbra in the subacute to chronic phase, especially reperfusion therapies are established presently (Figure 3). Considering the double-faceted roles of cells and the advantages of their pleiotropic mechanisms to tissue repair, we suggest that polarization strategies could be applied to restore brain parenchyma in pathological conditions, and that they are easily accessible therapeutic targets in ischemic stroke and neurological diseases.

  1. Conclusions

The administration of non-NSCs, such as BM-MSCs and BM-MNCs, is under clinical trial and it is unclear whether it will show any therapeutic improvement. A large number of clinical trials should be performed to confirm the real therapeutic effects of these treatments. The notion of cell polarization may be a conceptual framework. Researchers should investigate how and when to polarize the protective phenotype. Although future translational and clinical studies are required to support the idea of cell polarization, polarized cell therapies are gaining attention in the treatment of stroke and neurological diseases because they seem to have a protective function.

Minor: language requires extensive editing. Reading the text should be more fluid, sentences are arranged in awkward style at times.

Reply:

Thank you for your comments. The review was grammatically edited before the first submission. English editing has now been performed by a native speaker.

Abstract: I guess the authors meant that “more than half of the patients do not achieve functional independence”

Reply:

Thank you for your comments. We have revised the sentence accordingly.

(Page 2, 3rd sentence in abstract)

“However, half of the patients cannot achieve functional independence after treatment.”

Round 2

Reviewer 1 Report

The paper has improved a lot compared to the previous one. The reader's understanding has increased. as pending improvements, it is necessary to include the article search methodology. It is also necessary to expand the conclusions, so that anyone who reads it can make decisions, based on the evidence.

Author Response

it is necessary to include the article search methodology.

Reply:

Thank you for your comments. Accordingly, we have added the method section as below.

  1. Methods

A literature review was performed using PubMed as well as the National Institutes of Health clinical trial database (Clinicaltrials.gov). We searched articles published between April 1997 and July 2020 using the search terms “cell therapy”, “stroke”, “neurological disease”, “stem cell”, “microglia”, “mononuclear cell”, “microglial polarization”, “macrophage polarization”, and “A1 and A2 astrocyte”, “N1 and N2 neutrophil”, and “T cell”.

It is also necessary to expand the conclusions, so that anyone who reads it can make decisions, based on the evidence.

Reply:

Thank you for your comments. Accordingly, we have revised and added sentences in conclusion.

Although the cell therapies using BM-MSCs and BM-MNCs are promising clinical applications, it is unclear whether it will show any therapeutic improvement.

However, a polarized protective phenotype of astrocytes, neutrophils, microglia, T cell, and PBMCs may be existed.

Reviewer 2 Report

The authors have revised the manuscript according to my points. The manuscript now reads well and organized. 

Author Response

The authors have revised the manuscript according to my points. The manuscript now reads well and organized.

Reply:

I really appreciate your valuable comments.